# Fatty Acid Amides Suppress Proliferation via Cannabinoid Receptors and Promote the Apoptosis of C6 Glioma Cells in Association with Akt Signaling Pathway Inhibition

**DOI:** 10.3390/ph17070873

**Published:** 2024-07-02

**Authors:** Nágila Monteiro da Silva, Izabella Carla Silva Lopes, Adan Jesus Galué-Parra, Irlon Maciel Ferreira, Chubert Bernardo Castro de Sena, Edilene Oliveira da Silva, Barbarella de Matos Macchi, Fábio Rodrigues de Oliveira, José Luiz Martins do Nascimento

**Affiliations:** 1Programa de Pós-Graduação em Neurociências e Biologia Celular, Instituto de Ciências Biológicas, Universidade Federal do Pará, Belém 66075-110, Brazil; nagilamonteiro.s@gmail.com (N.M.d.S.); izabellacslopes@gmail.com (I.C.S.L.); edilene@ufpa.br (E.O.d.S.); 2Laboratorio de Neuroquímica Molecular e Celular, Instituto de Ciências Biológicas, Universidade Federal do Pará, Belém 66075-110, Brazil; barbarella@ufpa.br; 3Laboratório de Biologia Estrutural, Instituto de Ciências Biológicas, Universidade Federal do Pará, Belém 66075-750, Brazil; adangalue@gmail.com (A.J.G.-P.); chubert@ufpa.br (C.B.C.d.S.); 4Laboratório de Biocatálise e Síntese Orgânica Aplicada, Departamento de Ciências Exatas e Tecnológicas, Universidade Federal do Amapá, Macapá 68902-280, Brazil; irlon.ferreira@gmail.com; 5Programa de Pós-Graduação em Ciências Farmacêuticas, Departamento de Ciências Biológicas e da Saúde, Universidade Federal do Amapá, Macapá 68902-280, Brazil; 6Instituto Nacional de Ciência e Tecnologia em Neuroimunomodulação (INCT-NIM), Rio de Janeiro 21040-900, Brazil; 7Instituto Nacional de Ciência e Tecnologia de Biologia Estrutural e Bioimagem (INCT-INBEB), Rio de Janeiro 21941-902, Brazil; 8Programa de Pós-Graduação em Farmacologia e Bioquímica, Instituto de Ciências Biológicas, Universidade Federal do Pará, Belém 66075-110, Brazil; 9Laboratório de Controle de Qualidade e Bromatologia, Curso de Farmácia, Departamento de Ciências Biológicas e da Saúde, Universidade Federal do Amapá, Macapá 68902-280, Brazil; oliveirafabio.fr@gmail.com

**Keywords:** C6 cells, *Carapa guianensis* Aublet, fatty acid amides, endocannabinoids, FAA1 and FAA2, glioma

## Abstract

A glioma is a type of tumor that acts on the Central Nervous System (CNS) in a highly aggressive manner. Gliomas can occasionally be inaccurately diagnosed and treatments have low efficacy, meaning that patients exhibit a survival of less than one year after diagnosis. Due to factors such as intratumoral cell variability, inefficient chemotherapy drugs, adaptive resistance development to drugs and tumor recurrence after resection, the search continues for new drugs that can inhibit glioma cell growth. As such, analogues of endocannabinoids, such as fatty acid amides (FAAs), represent interesting alternatives for inhibiting tumor growth, since FAAs can modulate several metabolic pathways linked to cancer and, thus, may hold potential for managing glioblastoma. The aim of this study was to investigate the in vitro effects of two fatty ethanolamides (FAA1 and FAA2), synthetized via direct amidation from andiroba oil (*Carapa guianensis* Aublet), on C6 glioma cells. FAA1 and FAA2 reduced C6 cell viability, proliferation and migratory potential in a dose-dependent manner and were not toxic to normal retina glial cells. Both FAAs caused apoptotic cell death through the loss of mitochondrial integrity (ΔΨm), probably by activating cannabinoid receptors, and inhibiting the PI3K/Akt pathway. In conclusion, FAAs derived from natural products may have the potential to treat glioma-type brain cancer.

## 1. Introduction

Gliomas are rare cancers that affect the Central Nervous System (CNS); they are more prevalent in adults and occur more frequently in the elderly of age 64 and over [1]. Although only 1.4% of the world’s cancer types are gliomas, these tumors are highly aggressive, and patient survival time is less than a year after diagnosis [2,3]. Despite the different strategies currently used for treating gliomas, such as combinations of surgery resection, radiotherapy and chemotherapy, these therapies are not effective and frequently cause cancer drug resistance and recurrence after tumor resection [4,5]. Additionally, some drugs used for treating gliomas need to be administered at very high doses to enable them to cross the blood–brain barrier, thereby causing toxicity in peripheral tissues [6,7]. Thus, new therapeutic approaches must be developed, with a focus on killing glioma cells. Molecules capable of interacting with the endocannabinoids, which modulate several systems of the human organism including the CNS, have been investigated for the treatment of these malignant cells [8]. The current literature shows that the endocannabinoid system is related to the cancer development pathway. Its ligands act as antitumor agents in multiple cancer types, such as lung cancer, breast cancer, melanoma, neuroblastoma and glioma [9,10,11,12].

Both CB1 and CB2 endocannabinoid receptors have been commonly found in glioma cell lines and tumor biopsies [9,13]. CB2 is overexpressed in all astrocytomas, including gliomas, and levels are correlated with tumor malignancy grade [14].

Furthermore, previous studies have shown that endocannabinoid receptor ligands have anticancer efficacy against gliomas [15,16,17,18]. The most well-studied phytocannabinoids are cannabidiol (CBD) and delta-9-Tetrahydrocannabinol (∆^9^-THC), both extracted from *Cannabis sativa* [19]. However, *Cannabis* and its derivates are still illegal in most countries, which is a significant obstacle to using the plant as an anticancer agent. From this perspective, natural product derivates that are endocannabinoid-like are potential alternatives. Fatty acid amides are molecules that may be particularly suitable for substituting cannabinoids, due to their similar chemical structure [20,21].

Due to the abundance of plants in the Amazon region, this region represents a source of oils that can be used as the raw material for the synthesis of such molecules. *Carapa guianensis* Aublet, popularly known as Andiroba, is used in folk medicine for treating many diseases, including gastric cancer [22,23,24,25]. Andiroba oil samples (AO1 and AO2) derived from these plants contain a high amount of saturated and unsaturated fatty acids, such as palmitic, oleic and linoleic acids, which can be used for fatty acid amide synthesis [26,27,28].

The main aim of the present study was to test the anti-cancer effects of two groups of fatty ethanolamides, FAA1 and FAA2, in glioma cells. FAA1 and FAA2, which are considered part of the endocannabinoid system, were synthesized by biocatalysis from *C. guianensis* oil (AO1 and AO2), and have previously been characterized [29]. Here, we report data to show that the FAAs tested can reduce the proliferation and cell viability of glioma cells within a 12 h period, in a dose-dependent manner, and induce apoptosis in the cells via cannabinoid receptors. We also demonstrated an inhibitory effect of FAAs on active, phosphorylated Akt levels, along with mitochondrial transmembrane potential (ΔΨm) loss.

## 2. Results

### 2.1. Spectroscopic Analysis

AO1 and AO2 presented promising profiles regarding the main fatty acids identified. These results showed that temperature, solar radiation, precipitation and humidity may be responsible for influencing the fatty acid content of the seeds of *C. guianensis*. While AO1 mostly contained saturated fatty acids (41.5%) such as palmitic and stearic acid, AO2 only had a saturated fatty acid proportion of 25.6%. It is noteworthy that the monounsaturated fatty acid oleic acid (C18:1, ʊ-9) was identified as the largest constituent in both samples, accounting for 50.6% of all fatty acids present in AO1 and 40.2% in AO2 (Figure 1).

After the direct amidation reaction of the andiroba oils (AO1 and AO2) and ethalonamine, the formation of a group of fatty amides was confirmed by GC-MS and ^1^H and ^13^C NMR analysis. As expected, because the alkyl chains within the starting solutions do not suffer from structural alteration, ethalonamides were unambiguous and highly similar to the NMR spectrum signals from AO1 and AO2, respectively, which were in natura. The pure AO2 spectrum had as its main characteristic dds (double doublets) in the region of σ 4.30 ppm and 4.15 ppm, corresponding to glycerolic methylene protons, and a similar sign was found in AO1 (^1^H-NMR spectrum, see Appendix A). After the direct amidation reaction, the spectrum referring to FAA2 was characterized by a t (triplet) at 3.68 ppm (J = 3.3 Hz) and a q (quartet) at 3.38 ppm (J = 5.3 Hz), characteristic of the ethanolamide ethylene protons (-NH_2_-CH_2_-CH_3_-OH), in addition to the presence of glycerolic methylene protons, confirming the formation of an amide bond.

Based on the signal aspect, the regions between 60 and 70 ppm in the ^13^C-NMR spectrum exhibited signals associated with alpha-methylene triacylglycerol groups, more specifically the 68.88 and 62.10 ppm signals (AO1 and AO2) (Figure 2), while the proof of amide link formation is related to the -CH_2_ carbon from ethalonamide at 62.07 ppm (-CH_2_-OH) and 42.35 (-NH-CH_2_-) (Figure 2B). Furthermore, the group of signals close to 173 ppm correspond to the C=O link, which is characteristic of carboxyl. Moreover, the region between 130 and 127 ppm is related to unsaturated carbon (sp2) from unsaturated and polyunsaturated fatty acids that are present in the oil in both the fatty chains and amides and finally in the bigger magnetic field region (30–20 ppm), which are related to alkil carbons (-CH_2_-).

### 2.2. Treatment with FAAs Inhibits C6 Glioma Cell Growth

This is example 1 of an experiment: In order to evaluate the effects of FAAs on cell viability, C6 glioma cells were tested using the MTT assay for mitochondrial activity. Cells were treated with FAA1 (Figure 3A) or FAA2 (Figure 3B) at different concentrations (30, 60, 90 and 120 μg/mL) with incubation times ranging from 6 to 48 h. Both of the fatty acid amides significantly decreased glioma cell viability at all times analyzed and at all concentrations tested (*p* < 0.05) when compared with the control group. Furthermore, both FAAs displayed cytotoxicity in a dose-dependent manner, with greater cytotoxicity being observed at 6 h and 12 h. At 12 h, both FAAs caused significant reductions in cell viability, with the highest values being observed at 90 and 120 μg/mL (FAA1, 66.58% and 79.12%, respectively, and FAA2, 80.65% and 82.51%, respectively). The IC_50_ values for FAA1 (70 μg/mL) and FAA2 (30 μg/mL) were used in further experiments.

In addition, we performed an MTT cytotoxicity assay to evaluate whether or not the viability of non-malignant glial cells is affected by treatment with FAA1 and FAA2 (Figure 4). We observed that neither of the FAAs significantly affected the cells when compared to the control group.

### 2.3. FAAs Induce Apoptosis in C6 Cells

Double staining with annexin V-FITC/PI was used to determine whether the C6 cells underwent death by apoptosis or by necrosis when exposed to FAAs (Figure 5). The untreated group demonstrated a cell viability of 87.73% (calculated as a percentage of the total cell population), whereas the majority of cells exposed to 70 μg/mL FAA1 exhibited cell death that was classified as early apoptosis (8.37%), late apoptosis (15.33%) or necrosis (8%). A high death rate was also seen for cells incubated with FAA2 (30 μg/mL), where cells could be divided into early apoptosis (60.76%), late apoptosis (19.12%) and necrosis (0.07%). FAA2 had a more significant effect since it induced a higher early apoptosis rate compared to that of the FAA1 group at a lower concentration.

### 2.4. FAAs Trigger Mitochondrial Integrity (ΔΨm) Damage in Glioma Cells

We investigated mitochondrial integrity (ΔΨm) using the JC-1 probe and flow cytometry. As expected, the graph presented (Figure 6) shows high red (PE-A)/green (Alexa fluor 488-A) ratio levels for the control group, indicating mitochondrial transmembrane integrity. On the other hand, the C6 cells demonstrated low red (PE-A)/green (Alexa fluor 488-A) ratio levels when treated with either of the FAAs (IC_50_), and demonstrated significantly fewer JC-1 aggregates in comparison with the control group. The lower fluorescence ratio of the FAA-treated groups was similar to that of the positive control treated with carbonyl cyanide m-chlorophenylhydra-zone (CCCP), indicating mitochondrial damage. However, although FAA1 and FAA2 slightly affected the fluorescence intensity ratio, this was not considered significant.

In addition to the JC-1 probes, we also used a fluorescent green dye (MitoTracker) to investigate mitochondrial membrane potential (ΔΨm) alterations. Viable control cells (Figure 7, upper line) exhibited a higher fluorescence, indicating functional mitochondria that were capable of metabolizing the Mitotracker probe. In contrast, the FAA1- and FAA2-treated groups (Figure 7, middle and lower lines) presented a less intense green stain, indicating mitochondrial permeabilization due to FAA treatment.

### 2.5. FAAs Cell Cycle Analysis

Next, we explored the effects of the FAAs on the cell cycle. Cells were evaluated by flow cytometry, utilizing the PI chromosome counterstain. The treatment of C6 cells with FAAs for 12 h altered their cell cycle. Similarly to the positive control (Figure 8C), the cells that were treated with FAA1 or FAA2 presented lower percentages of cells in the S, G2 and M phases when compared with control cells (Figure 8A,B). The percentage of cells in the subG1 phase increased after treatment with FAA1 (26.58%) or FAA2 (19.38%) when compared with the control (1.79%), showing once more the presence of apoptotic cells, but not that of cell cycle arrest (Figure 8).

### 2.6. CB1 and CB2 Antagonist Analysis

In order to better clarify the mechanism of action of FAA1 and FAA2, we used specific antagonists, LY3210135 (CB1) and AM630 (CB2), to block these endocannabinoid receptors. The maximum FAA1 concentration (120 µg/mL) tested in this work reduced C6 viability to 17.34%. The addition of the CB1 antagonist LY320135 (10 µM) increased the cell viability to nearly 50%. The addition of the CB2 antagonist, AM630 (10 µM), also significantly increased the cell viability (50%). There was an additive effect when both inhibitors were added, suggesting that FAA1 is able to activate both receptors. After treatment with FAA2 (120 µg/mL), the cells showed a mean reduced cell viability of 13.74%. Pretreatment with LY3200135 significantly increased the cell viability to approximately 40%. The addition of the CB2 antagonist AM630 (10 µM) increased cell viability to approximately 80%. When both inhibitors were added, there was no additive effect, suggesting that FAA2 acts mainly via CB2 receptors. The results shown in Figure 9 indicate that FAA1 acts via CB1 and CB2, while FAA2 may act mainly via CB2.

### 2.7. FAA1 and FAA2 Induce Apoptosis in Glioma Cells through Akt Signalling Pathway Suppression

In order to better study the cellular mechanisms that lead to the loss of cell viability, we performed Western blotting. For this assay, whole-cell protein extracts from cells that were pre-treated with FAAs, or untreated, were labelled with Akt and pAkt antibodies. The Akt levels remained similar when comparing the control with the FAA-treated samples. The pAkt levels decreased significantly following treatments with the FAAs, as can be observed in Figure 10, indicating that the effects of these endocannabinoid-like molecules are mediated through the PI3K/Akt pathway.

## 3. Discussion

Previous studies have reported on the promising antineoplastic therapeutic potential of endocannabinoid system ligands [30,31,32]. In recent years, plant-derived fatty acid amides have been produced and employed as endocannabinoid-like endogenous signaling molecules to modulate cancer-related metabolic pathways [31]. Both of the fatty acid amides used in the present work were produced from andiroba oil fatty acids, which belong to a large family of signaling lipids called N-acylethanolamines (NAEs) and interact with the endocannabinoid system [33], and have already been studied by our research group, showing an effect on the Central Nervous System [29]. A number of studies have shown the ability of *Carapa guianensis* and its derivatives to inhibit different types of cancer [24,34,35], although no studies to date have reported any action of these molecules in brain cancers. Here, we demonstrate, for the first time, the antineoplastic effects of two FAAs derived from *Carapa guianensis* in C6 glioma cells. These substances reduced cell viability in a dose-dependent manner, activating an apoptosis pathway, which could be mediated by cannabinoid receptors, directly or indirectly.

FAA1 and FAA2 were both able to reduce the proliferation, viability and invasive properties of rat C6 glioma cells. They strongly decreased rat C6 cell viability in a concentration-dependent manner, with IC_50_ values of 70 μg/mL and 30 μg/mL, respectively, for FAA1 and FAA2, after 12 h of incubation. Other endocannabinoid and endocannabinoid-like molecules have also been shown to induce apoptosis in brain cancer cells. AEA, 2-AG and stearoylethanolamide (SEA) induce apoptosis in C6 glioma cells [36], and oleamide (OEA) induces the apoptosis of RG2 glioblastoma cells [37]. Some of these molecules are present in the FAAs tested here, namely SEA (11.1% in FAA1 and 5.5% in FAA2) and OEA (52.6% in FAA1 and 50.1% in FAA2) [29].

The effects of the treatment of cells with FAA1 and FAA2 are reflected in-cell morphology changes. The cells presented alterations that were characteristic of apoptosis, such as the formation of apoptotic bodies, vacuolization of the cytoplasm and chromatin condensation. These findings are similar to those observed when treating primary cultures of glioblastoma with the synthetic cannabinoids, WIN55, 212-2 and JWH1, for 48 h [38].

In addition, the FAAs were not cytotoxic to normal glial cells, in agreement with previous studies indicating that endocannabinoid receptors are more highly expressed in malignant tissues [39,40], and highlighting their potential for targeting tumor cells without affecting healthy cells. Previous studies have demonstrated that the administration of CB1 agonists to several cancer cell lineages (U343MG (U343), U87MG (U87), U251MG (U251) and T98G (T98)) also induced apoptotic cell death [38,40]. Similarly, co-incubation of U87MG and MZC cells with CBD (10 μM, 1 day) phytocannabinoid and the most commonly used glioma drug, TMZ (temozolomide) (400 μM, 1 day), significantly increased apoptosis [41]. Furthermore, other studies have reported decreased human glioma cell (U251) viability due to apoptosis following AEA treatment (10 μM, 24 h) [42]. As such, the FAAs studied herein were found to act in a similar manner to those of other endocannabinoid system ligands that have been previously tested in glioma cells.

We observed increased mitochondrial membrane depolarization in apoptotic glioma cells that were treated with FAA1 and FAA2; similar observations have also been reported in human monocytes treated with 16 μM CBD [43], and in C6 glioma cells treated with the synthetic cannabinoid, WIN 55,212-2 [44]. In a more recent study, a reduction in oxygen consumption and mitochondrial inhibition were observed in cells treated with CBD, confirming that mitochondrial alterations induced by cannabinoids and their analogues can activate apoptotic pathways [45]. The pro apoptotic effects of FAA1 and FAA2 were confirmed by Annexin V-FITC/PI double staining, which detects apoptosis and necrosis. C6 cells treated with the FAA1 and FAA2 demonstrated dose-dependent apoptotic induction rates. Differences in late apoptosis rates versus viable cells were also observed according to the stage of apoptosis. These results confirm that the presumable mechanism of cell death induced by FAA1 and FAA2 is apoptosis.

Using antagonists of CB1 and CB2 individually did not significantly protect the glioma cells from death due to FAA1. However, a mixture of LY3210135 and AM630 prevented cell toxicity, indicating the FAA1 amide pool targets CB1 and CB2 in a similar way. Conversely, the effect of FAA2 was significantly inhibited only by the CB2 antagonist (AM630), as well as by the CB1 and CB2 antagonists together, showing that the FAA2 pool preferably binds to CB2, which may be explained by the fact that CB2 is expressed more than CB1 in gliomas [14].

It is well known that the serine/threonine kinase protein kinase B (PKB)/Akt pathway plays a key role in cellular processes, such as cell proliferation, apoptosis, transcription and cell migration [46]. The AKT pathway is important in the genesis of several types of cancer, being overexpressed and playing critical roles in the survival, proliferation, invasion and migration of cancer cells. The phosphatidylinositol 3-kinase/AKT pathway regulates the traffic of ceramide in gliomas, a link between lipid signaling pathways involved in the control of cells survival, suggesting that there may be mutually inhibitory crosstalk between these two pathways. This crosstalk represents an important point in the signaling of malignant growth, prevention of apoptosis and promotion of invasion [47,48,49]. In the present study, Western blotting immunostaining showed that pAkt is reduced compared to Akt in cells treated with FAA1 and FAA2, suggesting that these FAAs are able to prevent the apoptosis evasion process that commonly develops in cancers for tumor maintenance. Likewise, other studies have shown the same reduction in Akt phosphorylation in human glioma and neuroblastoma lines and murine astrocytes, when using a synthetic cannabinoid (WIN 55,212-2) and CBD [38,41,48].

Taking our findings into consideration, together with those of previous reports, we believe that FAAs induce glioma cell death through a mitochondria-dependent apoptotic pathway [50,51]. Our results indicate that FAAs activate CB1 and CB2 receptors, which leads to ceramide accumulation, and later downregulation of the AKT/PKI pathway, probably activating pro-apoptotic proteins such as BAD. The phosphorylation of BAD causes it to translocate to the mitochondria, resulting in mitochondrial membrane integrity loss, which we also observed in our findings [52].

Observing the differences in the FAAs compositions, we noticed that the content of the predominant molecule (oleoylethanolamide) is close to 50% in both FAAs. Besides oleoylethanolamide, palmitoylethanolamide represents 26% of FAA1, while linoleoylethanolamide makes up 30.2% of FAA2. These differences in composition may justify why FAA2 seems to preferably activate CB2. However, according to Di Marzo and Picitelli [53] endocannabinoid-like mediators, such as palmitoylethanolamide and linoleoylethanolamide, only influence CB1 and CB2 indirectly. Other studies have reported that linoleoylethanolamide and oleoylethanolamide inhibit fatty acid amide hydrolase, which is responsible for degrading endocanabinoids, such as anandamide (AEA). In general, endocannabinoids are produced and degraded on demand, meaning that FAAH inhibition allows endocannabinoids to be available to activate the cannabinoid receptors for a longer time.

Thus, we found that fatty acid amides from *Carapa guianensis* exert positive proapoptotic effects in C6 glioma cells that are associated with the activation (directly or indirectly) of cannabinoid receptors, decreasing PI3K/Akt pathway signaling. Furthermore, it should be noted that the FAAs may also interact with other non-canonical receptors, such as PPARg (peroxisome proliferators activated receptor) and TRPV1 and 2 (Transient Receptor Potential Vanilloid type-1 and 2), which may play important roles in glioma pathophysiology and represent therapeutic targets [54,55]. We also believe that PI3K/Akt and BAD protein inhibition leads to mitochondrial transmembrane integrity loss and Cytochrome C release, thereby activating a caspase-dependent apoptotic pathway.

We demonstrate that fatty acid amides suppress proliferation in glioma cells. However, this study has limitations as it was only conducted in one lineage of cells, but this lineage has long been used as a glioma model for drug testing because of the good correlation between cell lineage and cell fate when drugs are tested. We have obtained initial data suggesting that these amides have similar effects in human lineage. Furthermore, we plan to investigate the impact of these amides in vivo animal studies.

## 4. Materials and Methods

### 4.1. Cell Culture

The C6 glioma cell line (CCL-107™, ATCC) used in this study was maintained in Dulbecco’s Modified Eagle’s Medium (DMEM), supplemented with 10% fetal bovine serum and penicillin–streptomycin (10 U/mL).

Primary glial cells were obtained from *Gallus gallus domesticus* embryos eight days (E8) after fertilization. Retina from embryos were harvested by manual dissection in CMF (Calcium-Free Medium). Briefly, retina tissue was enzymatically dissociated using 0.05% trypsin for 10 min. Using successive pipetting, cells were disaggregated and cultivated for 10 days in DMEM. All cells were maintained at 5% CO_2_ and 37 °C in a humidified incubator.

### 4.2. Transesterification of AO1 and AO2

The production of fatty acid amides (FAA1 and FAA2) from AO1 and AO2 was measured in March and in June in the same place, respectively. The samples were performed in 3 mL Erlenmeyer flasks, containing 150 mg (154 µL) oil, 475 µL ethanol and 10% of CALB (m/m of the oil). The mixtures were incubated at room temperature (32 °C) on a 130 rpm orbital shaker (Lucadema, São José do Rio Preto, Brazil). After 24 h, the reaction was completed, the reaction mixture was filtered and the organic phase was dried over sodium sulfate. The solvent was then removed under reduced pressure and the product was purified by gel silica column chromatography with a mixture of n-hexane–ethyl acetate (9:1) as an eluent and characterized using gas chromatography–mass spectrometry (GC-MS) analysis (Figure 1).

### 4.3. Process for Obtaining Fatty Ethanolamides by Lipase from Candida Antarctica-B (CAL-B)

The direct amidation reaction was performed following the methodology described by Oliveira et al. [29]. Briefly, andiroba oil (3 mL), AO1 and AO2 separately, ethanolamine (9 mL) and CAL-B (150 mg) were added to an Erlenmeyer flask (25 mL). The mixture was stirred for 24 h under controlled conditions (150 rpm at 35 °C), filtered for enzyme retention and washed with 15 mL hexane. Next, 15 mL of distilled water was added and the organic phase extracted with hexane (2 × 15 mL) in anhydrous sodium sulfate and again filtered. Finally, the reaction was purified by column chromatography on silica gel 60 using a hexane–ethyl acetate (9:1) mixture. The obtained amides (FAA1 and FAA2) were characterized by ^13^C nuclear magnetic resonance (NMR).

### 4.4. Analysis

#### 4.4.1. GC-MS Analysis

The peaks were detected in the GC-MS on a Shimadzu/GC 2010 apparatus coupled to a Shimadzu/AOC-5000 auto-injector and an electron beam impact detector (Shimadzu MS2010 Plus, Kyoto, Japan) (70 eV) equipped with a DB-5MS fused silica column (30 m × 0.25 mm × 0.25 mm) (65 kPa). The parameters used were 1:15 split ratio, helium as the drag gas, 1.0 mL injection volume, injector temperature of 250 °C, detector temperature of 270 °C, initial column temperature of 100 °C for 2 min, heating rate of 6 °C min^−1^ until 280 °C for 5 min. The total analysis time was 37 min. Identification of the fatty acid ethyl esters was carried out through a comparison of the fragmentation spectrum with those contained in the GC-MS library (MS database, NIST 5.0).

#### 4.4.2. Nuclear Magnetic Resonance (NMR) Analysis (^1^H and ^13^C)

The ^1^H and ^13^C NMR experiments used 500 MHz and 125 MHz spectrometers (Agilent Technologies Premium Shielded, Santa Clara, CA, USA), respectively. All samples (20 µL, approximately) were prepared by dissolving in 600 µL of deuterated chloroform (CDCl_3_, Cambridge Isotope Laboratories, Andover, MA, USA) and tetramethylsilane (TMS) as the internal standard. The chemical shifts were expressed in ppm.

### 4.5. Cell Viability Assay

Cell viability was assessed using the thiazolyl blue tetrazolium bromide (MTT) assay. Cells were seeded in 96-well plates, at a density of 1 × 10^4^/well, and incubated at 37 °C in 5% CO_2_. Cells were incubated with DMSO (0.3%) (vehicle control) or treated with FAA1 and FAA2, diluted to different concentrations, for specified periods of time. For the analysis of the effects of cannabinoid receptor, cells were incubated with the CB1 antagonist, 10 μM LY3210135, and/or the CB2 antagonist, AM630, 30 min before the treatment with fatty acid amides for 12 h; 120 µg/mL FAA1 and FAA2. After incubation, the medium was removed and replaced with serum-free medium containing 0.5 mg/mL MTT and incubated for 3 h at 37 °C. The MTT reagent was removed, the formazan crystals were solubilized using DMSO and the absorbance was recorded at 570 nm using a microplate spectrophotometer (Biorad Model 450 Microplate Reader, Hercules, CA, USA). Untreated cells were used as control.

### 4.6. Apoptosis Assay

To perform this analysis, cells were incubated with 70 μg/mL FAA1 or 30 μg/mL FAA2 for 12 h. Apoptosis was measured using the Annexin V-FITC Detection Kit (Invitrogen, Waltham, MA, USA). Following incubation, cells were detached with a 0.05% trypsin solution. Cell suspensions were then incubated with 10 μL Annexin V-FITC and 1 μL PI for 15 min, before being washed with PBS and suspended in 300 μL PBS. Cells that were treated with camptothecin (2 μM) were used as positive controls in apoptosis. The samples were analyzed using a BD FACSCanto II flow cytometer with BD FACSDiva software, 8.0 version (Franklin Lakes, NJ, USA), and a total of 10,000 events were collected for each sample and analyzed using the FlowWing software, 2.5.1 version (Turku, Finland). The mean fluorescence channel (MFC) was calculated and expressed ± standard error of the mean.

### 4.7. Analysis of Mitochondrial Transmembrane Potential (ΔΨm)

ΔΨm changes were determined using JC-1, a fluorescent carbocyanine dye (MitoProbeTM JC-1 Assay Kit; Invitrogen, Waltham, MA, USA). Cells were plated at a seeding density of 1 × 10^5^ cells/well in a 24-well plate. After 12 h of 70 μg/mL FAA1 or 30 μg/mL FAA2 treatment, the cells were harvested and washed with cold PBS. Subsequently, the cells were incubated with 50 μM JC-for 15 min at room temperature. The samples were analyzed using a BD FACSCanto II flow cytometer with BD FACSDiva software, 8.0 version (Franklin Lakes, NJ, USA) and a total of 10,000 events were collected for each sample and analyzed using the WinMDI 2.9 version (Joseph Trotter) software. The mean fluorescence channel (MFC) was calculated and expressed ± standard error of the mean.

### 4.8. Mitochondrial Labelling

C6 glioma cells (1 × 10^5^) were plated in 24-well plates before treating with 70 μg/mL FAA1 or 30 μg/mL FAA2. Cells were fixed with 3% Paraformaldehyde and later permeabilized using 1% Triton and incubated with 50 mM ammonium chloride (NH_4_Cl), followed by incubation with 200 nM of the MitoTracker Green (Invitrogen) probe. Cells were then mounted with ProLong™ Gold Antifade Mountant (Thermo Fisher Scientific, Waltham, MA, USA). Representative images were analyzed using an Axio Zeiss fluorescence microscope. As a positive control, cells were treated with 10 μM of carbonyl cyanide m-chlorophenylhydrazone (CCCP), an uncoupler of oxidative phosphorylation.

### 4.9. Cell Cycle Analysis

C6 cells were seeded in 12-well plates. Cells were incubated with 70 μg/mL FAA1 or 30 μg/mL FAA2 for 12 h. Cells were harvested by trypsinization and fixed in ice-cold 70% ethanol for at least one hour at 4 °C. The cells were centrifuged to remove ethanol and recentrifuged with cold PBS. The pellets were resuspended in a solution containing 100 μg/mL PI and 0.2 μg/mL RNase, and then incubated in the dark at room temperature for 45 min and protected from light. We used a BD FACSCanto II flow cytometer with BD FACS Diva software, 8.0 version (Franklin Lakes, NJ, USA) and a total of 10,000 events were collected for each sample and analyzed using the WinMDI 2.9 version (Joseph Trotter) software.

### 4.10. Western Blot Analysis

C6 cells were treated with 70 μg/mL FAA1 or 30 μg/mL FAA2 for 12h before total proteins were extracted from samples. Equal amounts (50 μg) of protein extracted from each condition were separated using SDS-PAGE (12% acrylamide), blotted on nitrocellulose membranes, blocked in 5% milk and then probed with an appropriate concentration of selected primary antibodies. The incubation with the primary antibodies Akt Rabbit Ab (Cell Signaling Technology, Danvers, MA, USA; 1:1000) and Phospho-Akt Rabbit Ab (Ser473) (Cell Signaling Technology; 1:1000) took place overnight at 4 °C. Subsequently, the membranes were washed with PBS and incubated for 1h with a secondary antibody, horseradish peroxidase (HRP) conjugated to anti-rabbit IgG antibody (Abcam, Cambridge, UK; 1:1000), for 1 h at room temperature. Detection was visualized using a chemiluminescence kit (Bio-Rad Laboratories, Hercules, CA, USA). The images were captured using an Amersham Imager 600 (GE Healthcare, Chicago, IL, USA). Quantitative analysis was performed using the Image J software (1.53e version) with β-actin as the endogenous control.

### 4.11. Statistical Analysis

All experiments were performed in triplicate. The mean and standard deviations of at least three experiments were determined. The results were analyzed by GraphPad Prism software, 6 version (GraphPad Software, Inc., Boston, MA, USA), using one-way analysis of variance (ANOVA) and Tukey’s post-test. The differences were considered significant when *p* < 0.05.

## 5. Conclusions

In conclusion, the data indicate that the fatty acid amides studied demonstrate a therapeutic potential to inhibit the growth of gliomas via Akt inhibition. As such, further investigations are needed to characterize the interactions of the FAAs at the receptor level, in order to elucidate the mechanism of action of these FAAs and eventually define the extent to which endocannabinoids may be able to contribute to effectively treating gliomas and improving patient survival.

## Figures and Tables

**Figure 1 pharmaceuticals-17-00873-f001:**
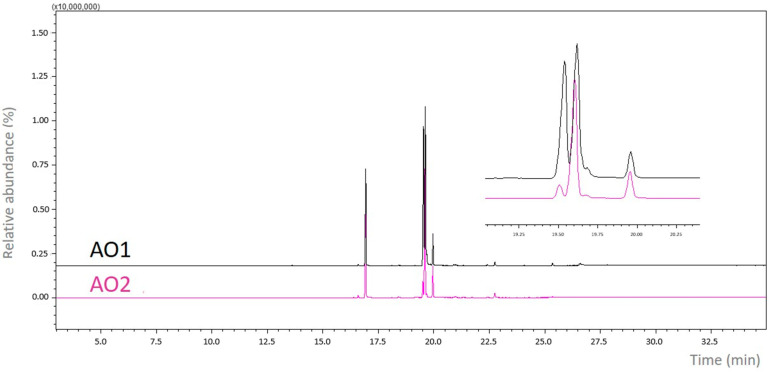
Chromatogram of the samples from AO1 and AO2 of *C. guianensis*.

**Figure 2 pharmaceuticals-17-00873-f002:**
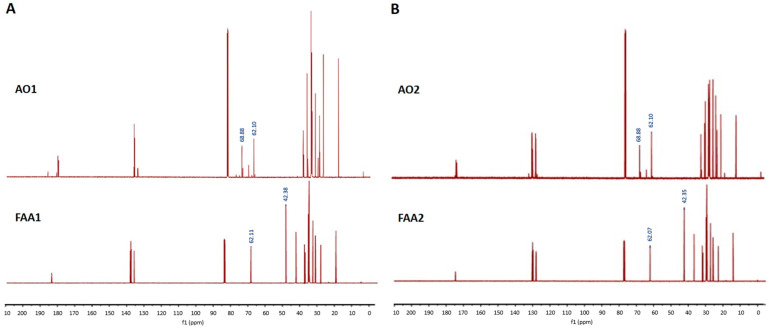
^13^C NMR spectra of the AO1 and FAA1 (**A**) and AO2 and FAA2 (**B**).

**Figure 3 pharmaceuticals-17-00873-f003:**
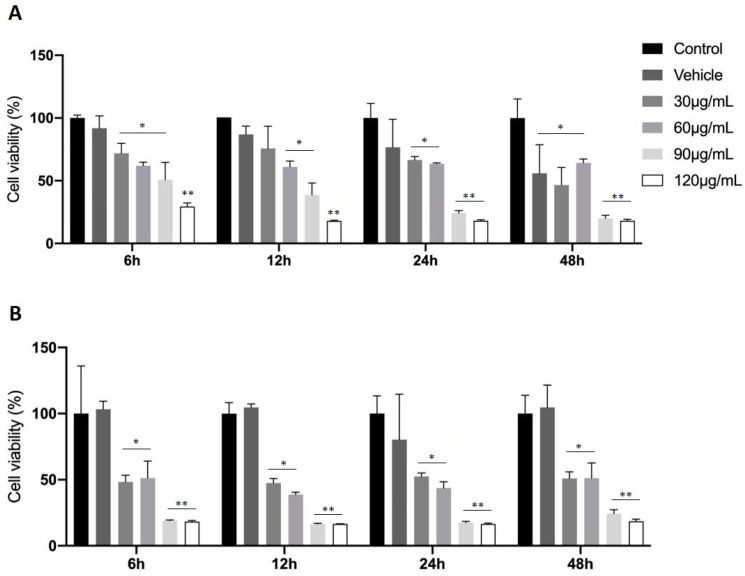
Effects of fatty acid amides (FAAs) on C6 cell viability, as measured by the MTT assay. C6 cells were treated with FAA1 (**A**) and FAA2 (**B**) for 6 h to 48 h, at concentrations ranging from 30 to 120 μg/mL. All concentrations reduced cell viability when compared to the control group. * *p* < 0.05 and ** *p* < 0.01 all groups vs. control (ANOVA, Tukey’s post-test).

**Figure 4 pharmaceuticals-17-00873-f004:**
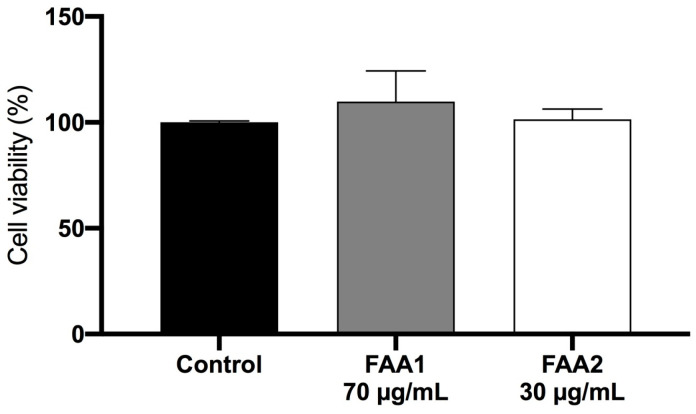
Measurement of glial cell viability (MTT assay) after 12 h incubations with IC_50_ concentrations of fatty acid amides; 70 µg/mL FAA1 and 30 µg/mL FAA2. FAA1 and 2 did not significantly affect cell viability, compared to the control group. Data are presented as the means ± SD of three independent experiments (ANOVA, Tukey’s post-test).

**Figure 5 pharmaceuticals-17-00873-f005:**
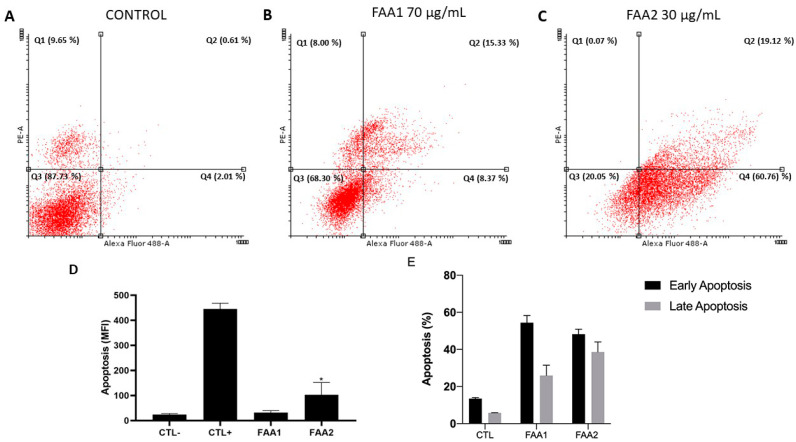
Treatment of C6 cells with FAAs induced apoptosis, as shown by the dot plot graphs for the Annexin V-FITC/PI staining assay using flow cytometry. Scatter plots show PE (*y*-axis) vs. Alexa Fluor (*x*-axis). (Q1) Upper left quadrant shows nonviable necrotic cells, (Q2) upper right quadrant represents late apoptotic cells, (Q3) viable cells are shown in the lower left quadrant, and (Q4) early apoptotic cells are in the lower right quadrant. (**A**) Untreated cells (control). (**B**,**C**) Cells treated with FAA1 and FAA2, respectively. Note that treatment of C6 cells with either of the FAAs for 12 h increased the percentage of early and late apoptotic cells in the population, compared to the control group. FAA2 induced a significantly higher early apoptosis rate, compared to that observed for FAA1. (**D**) Fluorescence intensity of cells labeled with Annexin V. (**E**) Quantification of the percentage of early and late apoptotic C6 cells. CTL−: Untreated cells, CTL+: Cells treated with camptothecin (2 μM). Data are presented as means ± SD of three independent experiments. * *p* < 0.05 vs. CTL−.

**Figure 6 pharmaceuticals-17-00873-f006:**
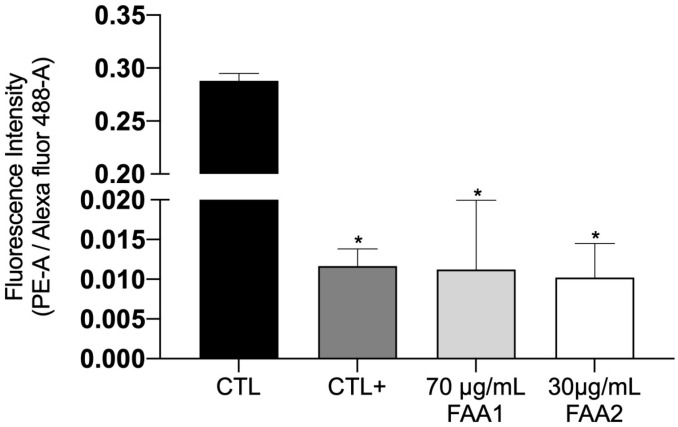
Measurement of mitochondrial membrane integrity (ΔΨm) disruption using JC-1 probe analysis with flow cytometry. The incubation of C6 cells with both of the FAAs (12 h) reduced the red (PE-A)/green (Alexa fluor 488-A) fluorescence ratio, compared to the control group. CTL: Untreated cells. CTL+: Cells treated with 10 μM CCCP. Data represent means ± SD of three independent experiments. * *p* < 0.001 vs. control group.

**Figure 7 pharmaceuticals-17-00873-f007:**
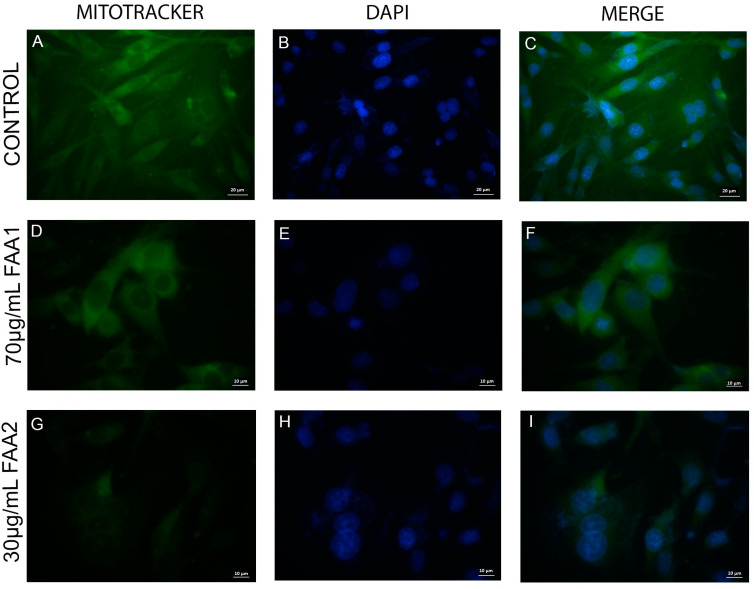
Representative micrographs of fluorescence in cells labelled with DAPI (ProLong™ Gold Antifade Mountant with DAPI—blue color) and Mitotracker green probes (green color). Control (**A**–**C**), 70 µg/mL FAA1 (**D**–**F**) and 30 µg/mL FAA2 (**G**–**I**) are shown with individual fluorescent stains or in overlapping images. 30 µg/mL FAA2-treated cells (**G**) showed less green fluorescence, compared to 70 µg/mL FAA1 (**D**) and control group (**A**) after 12 h of treatment. Representative images were analyzed using an Axio Zeiss fluorescence microscope; white arrows indicate nuclear fragmentation and formation of apoptotic bodies. Bar scales: 10 μm (**D**–**I**) and 20 μm (**A**–**C**).

**Figure 8 pharmaceuticals-17-00873-f008:**
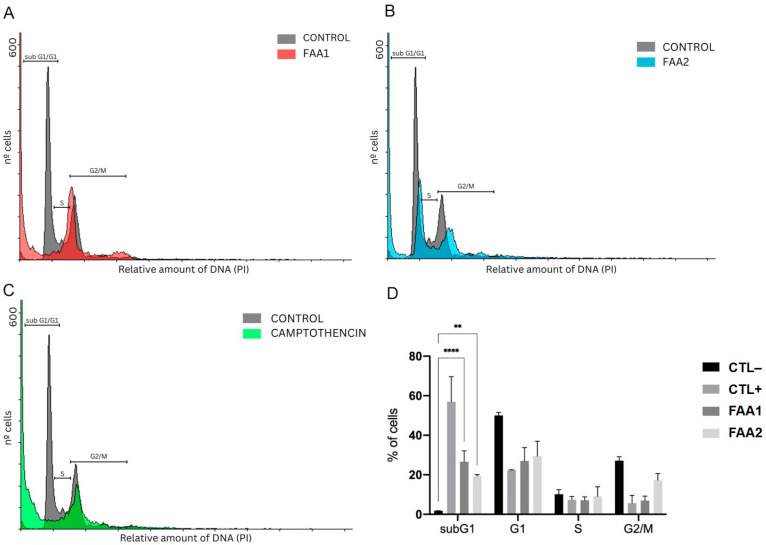
Treatment of cells with either FAA1 or FAA2 increased the percentage of cells in sub G1/G1, indicating statistically significant apoptosis. C6 cells were treated with both FAAs groups at IC_50_ concentrations; 70 µg/mL FAA1 (**A**) and 30 µg/mL FAA2 (**B**), except for the positive control (2 µM Camptothecin) (**C**). (**D**) Effects of both FAAs and Camptothecin treatments on cell cycle distribution of C6 glioma cells. Data are presented as the mean ± SEM of three independent experiments. ** *p* < 0.01 vs. CTL− and **** *p* < 0.001 vs. CTL−.

**Figure 9 pharmaceuticals-17-00873-f009:**
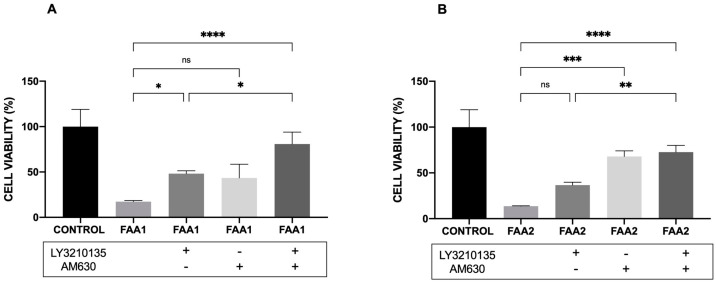
Effects of CB receptor antagonists, alone and in combination, on the antiproliferative effects of FAAs (FAA1 and FAA21) in C6 glioma cells. (**A**) Cells treated with the CB1 or CB2 antagonists, or both, followed by FAA1 incubation. (**B**) Cells treated with CB1 and CB2 antagonists, or both, followed by FAA2. Cells were incubated with inhibitors 30 min before fatty acid amides. Data are presented as the means ± SD of three independent experiments. * *p* < 0.05, ** *p* < 0.01, *** *p*< 0.001, **** *p* < 0.0001, ns (not significant) (ANOVA, Tukey’s post-test).

**Figure 10 pharmaceuticals-17-00873-f010:**
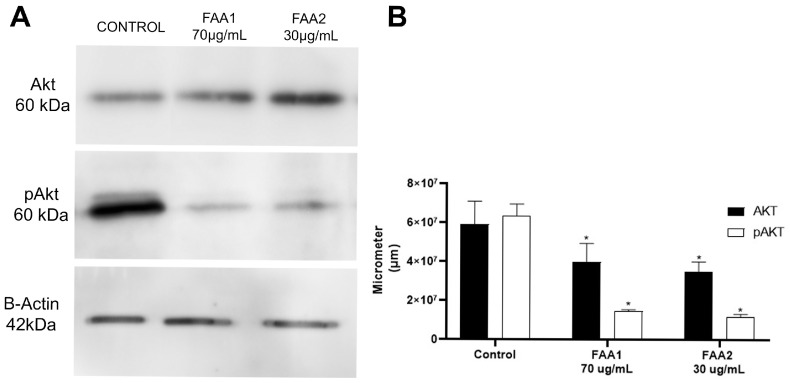
Treatment of C6 cells with 70 µg/mL FAA1 or 30 µg/mL FAA2 reduced Akt phosphorylation. Protein expression was detected by Western blotting (**A**) and bands areas were quantified using Image J software, 1.53e version (**B**). Data represent means ± SD of three independent experiments. * *p* < 0.001 vs. control group.

## Data Availability

The original contributions presented in the study are included in the article/Appendix A, further inquiries can be directed to the corresponding author.

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
