# Peer review of "Fatty Acid Amides Suppress Proliferation via Cannabinoid Receptors and Promote the Apoptosis of C6 Glioma Cells in Association with Akt Signaling Pathway Inhibition"

_pharmaceuticals, 2024, doi:10.3390/ph17070873_

Round 1

Reviewer 1 Report

Comments and Suggestions for Authors

1.      The C6 glioma cells were studied in this study. Authors must mention whether the FAA1 and FAA2 can cross BBB (brain-blood-barrier) into CNS. If FAAs does not enter CNS, authors must discuss how to use the FAAs to treat C6 glioma cells.

2.      As shown in figure 2, the components of AO1 and AO2 are similar to FAA1 and FAA2. Whether AO1 and AO2 are cytotoxic to C6 glioma cells?

3.      Due to 70ug/ml FAA1 and 30 ug//ml FAA2 were be used on normal glioma cells (Figure 4). Authors must provide the viability of C6 glioma cells treated with 70ug/ml FAA1 and 30 ug//ml FAA2.

4.      The paper used 2 uM camptothecin as positive control. Can authors provide the cytotoxic effect on 2 uM camptothecin-treated normal glioma cells?

5. Generally, cell cycle contains G0/G1, S and G2/M phase while sub-G1 is not involved in cell cycle phase. Sub-G1 usually indicated apoptotic cells. In this manuscript showed the FAAs increased the percentage of G1 and Sub-G1 and indicated the significant cell cycle arrest. However, only sub-G1 increased in figure 8, the G1, S and G2/M phase were decreased. This result did not indicate the cell cycle arrest by FAAs-treated. The result only indicated the apoptotic cell。 Please modify the content of manuscript.  

Author Response

We thank the reviewer comments towards our work and the opportunity to revise it and to make it clear.

  1. The C6 glioma cells were studied in this study. Authors must mention whether the FAA1 and FAA2 can cross BBB (brain-blood-barrier) into CNS. If FAAs does not enter CNS, authors must discuss how to use the FAAs to treat C6 glioma cells.

Reply: We understand the reviewer's concern. Our research group is currently working with fatty amides derived from vegetable oils and using models in the central nervous system. We have achieved positive results, as demonstrated in the study by Oliveira et al (2020). This study showed that the same fatty amides used in our work were effective in improving parameters in chemical seizure models induced by pentylenetetrazole in Swiss mice. To provide clarity, we have included this information in the discussion section, specifically on line 282-283.

Oliveira, F.R.; Rodrigues, K.E.; Hamoy, M.; Sarquis, I.R.; Hamoy, A.O.; Crespo-Lopez, M.E.; Ferreira, I.M.; Macchi, B.M.; do Nascimento, J.L.M. Fatty Acid Amides Synthesized from Andiroba Oil (Carapa Guianensis Aublet.) Exhibit Anticonvulsant Action with Modulation on GABA-A Receptor in Mice: A Putative Therapeutic Option. Pharmaceuticals 2020, 13(3), 43.

  1. As shown in figure 2, the components of AO1 and AO2 are similar to FAA1 and FAA2. Whether AO1 and AO2 are cytotoxic to C6 glioma cells?

Reply: We have also looked to the question asked by the reviewer regarding whether AO1 and AO2 are cytotoxic to C6 glioma cells. The experiment showed that andiroba oils AO1 and AO2, did not have impact on the viability of glioma cells (below). Additionally, there is no mention in the literature about the oil's activity in inhibiting brain cancer. There are significant differences in activity due to the presence of amide functional groups in the FAA structure.

Viability of Glioblastoma cells treated with Andiroba oil (72h) measured by the MTT assay. There was no significant difference between groups. Data are presented as the mean ± SEM of three independent trials. (ANOVA, Tukey’s post hoc test).

  1. Due to 70ug/ml FAA1 and 30 ug//ml FAA2 were be used on normal glioma cells (Figure 4). Authors must provide the viability of C6 glioma cells treated with 70ug/ml FAA1 and 30 ug//ml FAA2.

Reply: We thank the reviewer to call attention for that. The cytotoxicity test performed on glioma involved using FAA in concentrations ranging from 30 to 120 ug/mL. This was done to understand the cell viability pattern of these substances and to obtain the IC50 value. The IC50 values obtained were FAA1 (70 μg/mL) and FAA2 (30 μg/mL), respectively. These concentrations were selected to continue with all other experiments. Both concentrations were used to evaluate normal cells and demonstrate that there was no death-inducing effect, as shown in figure 4.

  1. The paper used 2 uM camptothecin as positive control. Can authors provide the cytotoxic effect on 2 uM camptothecin-treated normal glioma cells?

Reply: We have also looked to the interestingly question asked by the reviewer regarding the used 2uM camptothecin as positive control. For this experiment, we did not use camptothecin-treated normal  cells, as the literature indicates that high concentrations of this drug are necessary to induce apoptosis in normal neural cells, such as cortical cells and astrocytes, with average values of 2.7 and 4.7 μg/mL, respectively. For C6 cells, the IC50 value was, on average, 0.016 μg/mL. These results suggest that cell cycle regulation plays an important role in determining the effect of camptothecin on malignant and normal cells. Data from literature also indicates that a concentration of 2 µM of camptothecin is needed to induce apoptosis in order to observe all signs indicative of apoptosis in cancer cells.

Checa-Chavarria E, Rivero-Buceta E, Sanchez Martos MA, Martinez Navarrete G, Soto-Sánchez C, Botella P, Fernández E. Development of a Prodrug of Camptothecin for Enhanced Treatment of Glioblastoma Multiforme. Mol Pharm. 2021 Apr 5;18(4):1558-1572. doi: 10.1021/acs.molpharmaceut.0c00968. Epub 2021 Mar 1. PMID: 33645231; PMCID: PMC8482753.

Marciniak B, Kciuk M, Mujwar S, Sundaraj R, Bukowski K, Gruszka R. In Vitro and In Silico Investigation of BCI Anticancer Properties and Its Potential for Chemotherapy-Combined Treatments. Cancers (Basel). 2023 Sep 6;15(18):4442. doi: 10.3390/cancers15184442. PMID: 37760412; PMCID: PMC10526149.

  1. Generally, cell cycle contains G0/G1, S and G2/M phase while sub-G1 is not involved in cell cycle phase. Sub-G1 usually indicated apoptotic cells. In this manuscript showed the FAAs increased the percentage of G1 and Sub-G1 and indicated the significant cell cycle arrest. However, only sub-G1 increased in figure 8, the G1, S and G2/M phase were decreased. This result did not indicate the cell cycle arrest by FAAs-treated. The result only indicated the apoptotic cell。Please modify the content of manuscript.  

Reply:  We thank the reviewer suggestion and now we provide modifications in Figure 8.  The modifications have been made in the text as follows:

Line 223: The topic title has been changed to "2.5. FAAs Cell Cycle Analysis".

Lines 230-231: The final text of the paragraph has been modified to "...showing once more the presence of apoptotic cells, but not the cell cycle arrest." In addition to changes in the caption of Figure 8 (line 235) and changes to the discussion of the article in lines 328-331."

Reviewer 2 Report

Comments and Suggestions for Authors

This is a nice study on the effects of fatty acid amides on GBM cells from rats. The authors found an overall reduction of GBM cell aggressiveness induced by FAA1 and 2 endocannabinoid analogs derived from Carapa guianensis.  The mechanism of action is suggested to be through Pi3/Akt pathway, with activation of cannabinoid receptors. While the study has a lot of potential, some issues need to be assessed.

Major points: 

- Figure 3: stats: Please clarify if there is * missing  (comparisons with control)

- Figure 7: I do not agree with the authors on panel J. It is not clear that the arrows show apoptotic bodies or nuclear fragmentation. Please provide better examples. Also, if they want to show the nuclear changes they need to show a comparison with the control.

To complement claims in the mitochondrial role I would suggest including assays in mitochondrial structure (i.e. electron microscopy, or maybe cardiolipin content, )

- Figure 10, L 365-373. I would suggest adding some more insight into the pathway (i.e. mTOR status) and the apoptosis process (i.e. BAD, caspases WB), It will be interesting to see ceramide accumulation.

- Discussion. Could the authors comment on the limitations of the study? i.e. rat cell line (not human, no patient-derived, no in vivo experiments)

Minor points: 

- L89 Please, remove the template paragraph

- Figure 3 caption and panels don't match or letters are missing

- L158-159 Please, remove template sentences

- L389 Could the authors comment on the relevance of the date of collection? 

Author Response

We thank the reviewer comments towards our work and the opportunity to revise it and to make it clear.

This is a nice study on the effects of fatty acid amides on GBM cells from rats. The authors found an overall reduction of GBM cell aggressiveness induced by FAA1 and 2 endocannabinoid analogs derived from Carapa guianensis.  The mechanism of action is suggested to be through Pi3/Akt pathway, with activation of cannabinoid receptors. While the study has a lot of potential, some issues need to be assessed.

Major points: 

- Figure 3: stats: Please clarify if there is * missing (comparisons with control)

Reply:  We agree with reviewer.  The statistics have been revised and the figure has been replaced.

h- Figure 7: I do not agree with the authors on panel J. It is not clear that the arrows show apoptotic bodies or nuclear fragmentation. Please provide better examples. Also, if they want to show the nuclear changes they need to show a comparison with the control.

Reply: We agree with the reviewer and removed the sentence to make the text concise and understandable. The final text of the paragraph has been modified to the morphological examination showed that compared to the standard size, shape and elongated form of the C6 cells without treatment with those currently under treatment resulted in cell shrinkage and appearance of some round and dead cells.

To complement claims in the mitochondrial role I would suggest including assays in mitochondrial structure (i.e. electron microscopy, or maybe cardiolipin content, )

Reply: Reply: We thank the reviewer for bringing this to our attention, but we did have limitations in verifying other techniques that reveal structural changes in mitochondria. Currently, the two techniques used in this work - measurement of mitochondrial membrane integrity (ΔΨm) disruption using JC-1 probe analysis with flow cytometry and mitotracker probe - indicate mitochondrial permeabilization due to FAA treatment were sufficient to verify changes in mitochondria compared with the control.

- Figure 10, L 365-373. I would suggest adding some more insight into the pathway (i.e. mTOR status) and the apoptosis process (i.e. BAD, caspases WB), It will be interesting to see ceramide accumulation.

Reply: We are grateful for the reviewer comments on this interesting question. The AKT pathway is important in the genesis of several types of cancer, being overexpressed and playing critical roles in the survival, proliferation, invasion, and migration of cancer cells. Phosphatidylinositol 3-kinase/AKT pathway regulates the traffic of ceramide in gliomas, a link between lipid signaling pathways involved in the control of cells survival, suggesting that there may be mutually inhibitory crosstalk between these two pathways. This crosstalk represents an important point in the signaling of malignant growth, prevention of apoptosis and promotion of invasion.

- Discussion. Could the authors comment on the limitations of the study? i.e. rat cell line (not human, no patient-derived, no in vivo experiments)

Reply: We agree on the limitations of the study that could be made in more than one lineage, but we only have results in the lineage in C6. This lineage has long been used as a glioma model for drug testing, because of the good correlation between cell lineage and cell fate when drugs are tested. We have initial data suggesting that these amides have similar effects in human lineage as in C6 cells.  Furthermore, we plan to investigate the impact of these amides in vivo animal studies.

Minor points: 

- L89 Please, remove the template paragraph.

Reply: The sentence was removed.

- Figure 3 caption and panels don't match or letters are missing.

Reply: The caption was corrected.

- L158-159 Please, remove template sentences.

Reply: The sentences were removed.

- L389 Could the authors comment on the relevance of the date of collection? 

Reply: We appreciate the reviewer comments stating that the date of collection is not relevant. The date of collection has been removed from the text, but the month has been kept due to seasonality can influence the chemical composition of the oils.

Round 2

Reviewer 1 Report

Comments and Suggestions for Authors

Authors have made response to the major comments

Author Response

(1) Comment: h- Figure 7: I do not agree with the authors on panel J. It is not clear that the arrows show apoptotic bodies or nuclear fragmentation. Please provide better examples. Also, if they want to show the nuclear changes they need to show a comparison with the control.

Author Reply: We agree with the reviewer and removed the sentence to make the text concise and understandable. The final text of the paragraph has been modified to the morphological examination showed that compared to the standard size, shape and elongated form of the C6 cells without treatment with those currently under treatment resulted in cell shrinkage and appearance of some round and dead cells.

While the authors have agreed with this, they have not sufficiently addressed this comment. It is unclear what changes were made in the manuscript to address the comment. The paragraph in the text discussing Fig 7J (L205-208) and the caption for Fig 7J still describe the arrows as showing nuclear fragmentation and apoptotic bodies.

Reply: We agree with the reviewer remarks and decided to remove panel 7J and its caption because it was difficult to distinguish between apoptosis and nuclear fragmentation.

Lines 212-215: “Additionally, DAPI staining of C6 cells demonstrated changes in cell morphology. FAA2 treatment (30 μg/mL) induced structural signs of apoptosis, such as nuclear fragmentation and the formation of apoptotic bodies (Figure 7J), whereas 70 μg/mL FAA1 did not induce visible morphological changes.” - This sentence was removed.

Lines 221-222: the caption for panel 7J was removed – “(J) Nuclear changes typical of the apoptotic process in C6 cells after treatment with 30 µg/mL FAA2 for 12 h.

The figure 7 was replaced, without panel J.

(2) Comment: Figure 10, L 365-373. I would suggest adding some more insight into the pathway (i.e. mTOR status) and the apoptosis process (i.e. BAD, caspases WB), It will be interesting to see ceramide accumulation.

Author Reply: We are grateful for the reviewer comments on this interesting question. The AKT pathway is important in the genesis of several types of cancer, being overexpressed and playing critical roles in the survival, proliferation, invasion, and migration of cancer cells. Phosphatidylinositol 3-kinase/AKT pathway regulates the traffic of ceramide in gliomas, a link between lipid signaling pathways involved in the control of cells survival, suggesting that there may be mutually inhibitory crosstalk between these two pathways. This crosstalk represents an important point in the signaling of malignant growth, prevention of apoptosis and promotion of invasion.

Suggest including this in the introduction or discussion section of the paper in addition to the response

Reply: We also introduced a line in discussion as it reads.

Lines 345-346 (The Akt signaling pathway has been associated with apoptosis suppression pathways in glioma cells) were replaced by “The AKT pathway is important in the genesis of several types of cancer, being overexpressed and playing critical roles in the survival, proliferation, invasion, and migration of cancer cells. Phosphatidylinositol 3-kinase/AKT pathway regulates the traffic of ceramide in gliomas, a link between lipid signaling pathways involved in the control of cells survival, suggesting that there may be mutually inhibitory crosstalk between these two pathways. This crosstalk represents an important point in the signaling of malignant growth, prevention of apoptosis and promotion of invasion” (Lines 346-353).

(3) Comment: Discussion. Could the authors comment on the limitations of the study? i.e. rat cell line (not human, no patient-derived, no in vivo experiments)

Reply: We agree on the limitations of the study that could be made in more than one lineage, but we only have results in the lineage in C6. This lineage has long been used as a glioma model for drug testing, because of the good correlation between cell lineage and cell fate when drugs are tested. We have initial data suggesting that these amides have similar effects in human lineage as in C6 cells.  Furthermore, we plan to investigate the impact of these amides in vivo animal studies.

The authors should include the limitations in the discussion section of the paper in addition to the response.

Reply: We also introduced the sentence in discussion as it reads.

Lines (388-393): We demonstrate that fatty acid amides suppress proliferation in glioma cells. However, this study has limitations it was only conducted in one lineage of cells, but this lineage has long been used as a glioma model for drug testing, because of the good correlation between cell lineage and cell fate when drugs are tested. We have initial data suggesting that these amides have similar effects in human lineage.  Furthermore, we plan to investigate the impact of these amides in vivo animal studies.